

# Integrating multiple microarray dataset analysis and machine learning methods to reveal the key genes and regulatory mechanisms underlying human intervertebral disc degeneration

Hongze Chang, Xiaolong Yang, Kemin You, Mingwei Jiang, Feng Cai, Yan Zhang, Liang Liu, Hui Liu and Xiaodong Liu

Department of orthopedics, Shanghai Yangpu Hospital Affiliated to Tongji University, Shanghai, China

Corresponding author
Xiaodong Liu,
xiaodong.liu@tongji.edu.cn

## ABSTRACT

Intervertebral disc degeneration (IDD), a major cause of lower back pain, has multiple contributing factors including genetics, environment, age, and loading history. Bioinformatics analysis has been extensively used to identify diagnostic biomarkers and therapeutic targets for IDD diagnosis and treatment. However, multiple microarray dataset analysis and machine learning methods have not been integrated. In this study, we downloaded the mRNA, microRNA (miRNA), long noncoding RNA (lncRNA), and circular RNA (circRNA) expression profiles (GSE34095, GSE15227, GSE63492 GSE116726, GSE56081 and GSE67566) associated with IDD from the GEO database. Using differential expression analysis and recursive feature elimination, we extracted four optimal feature genes. We then used the support vector machine (SVM) to make a classification model with the four optimal feature genes. The ROC curve was used to evaluate the model's performance, and the expression profiles (GSE63492, GSE116726, GSE56081, and GSE67566) were used to construct a competitive endogenous RNA (ceRNA) regulatory network and explore the underlying mechanisms of the feature genes. We found that three miRNAs (hsa-miR-4728-5p, hsa-miR-5196-5p, and hsa-miR-185-5p) and three circRNAs (hsa_circRNA_100723, hsa_circRNA_104471, and hsa_circRNA_100750) were important regulators with more interactions than the other RNAs across the whole network. The expression level analysis of the three datasets revealed that BCAS4 and SCRG1 were key genes involved in IDD development. Ultimately, our study proposes a novel approach to determining reliable and effective targets in IDD diagnosis and treatment.

## INTRODUCTION

Lower back pain, commonly caused by intervertebral disc degeneration (IDD), can be a significant socioeconomic burden on patients (*Vergroesen et al., 2015*). IDD is characterized by the apoptosis of nucleus pulposus (NP) cells, the degradation of extracellular matrix

(ECM) components, and several contributing factors including genetics and environment (*Battie et al., 2008*; *Feng, Egan & Wang, 2016*). However, the precise etiology of IDD remains largely unknown. Diagnosing degenerative disc disease is difficult, and common IDD treatment and management strategies primarily consist of conservative management or surgical treatment to relieve pain, without resolving the underlying tissue pathology (*An et al., 2003*; *Zaina et al., 2016*). To identify potential biomarkers and specific therapeutic targets, a more detailed understanding of the molecular and cellular events underlying IDD formation is needed.

The etiology of IDD is complex, but over the past several decades, it has become clear that genetic factors are most dominant. Multiple candidate genes, including *thrombospondin-2*, vitamin D receptor, *COL2A1*, *ACAN*, interleukins (*IL1α, IL 1β*, and *IL6*), matrix metalloproteinases (*MMP-3* and *MMP-9*), and growth/differentiation factor 5, have been associated with the pathophysiological process of IDD development (*Feng, Egan & Wang, 2016*; *Kalb et al., 2012*; *Wang et al., 2018b*; *Yuan et al., 2018*). Genome-wide association studies (GWAS) have been used to help identify novel variants. Several GWAS found that the genetic polymorphisms of *PARK2* and *CHST3* were relevant to IDD etiology (*Song et al., 2013*; *Williams et al., 2013*). Additionally, some bioinformatics analyses based on gene expression profiles revealed that *FYN*, *PRKCD*, *YWHAB*, *YWHAZ*, *AR*, *Fibronectin 1*, *COL2A1*, *β-catenin*, *COL6A2*, *IBSP*, *RAP1A*, and *FOXF2* genes may play key roles in IDD development (*Chen et al., 2013*; *Guo et al., 2017*; *Ji et al., 2015*). Although a considerable number of genes associated with IDD development have been found, early IDD diagnosis and precise treatment remain difficult and require further study.

Integrated analyses using combined multiple microarray data can provide a more accurate understanding of the interplay across multi-level genomic features and the molecular mechanisms that cause complex diseases (*Momtaz et al., 2018*). Machine learning is a type of artificial intelligence that can "learn" a model using past data in order to predict future data. Machine learning algorithms have been used in key feature training, recognition, and group classification (*Huang et al., 2018*). Modern researchers have unprecedented access to machine learning methods that can elucidate complex molecular mechanisms and predict disease genes from large biomedical datasets (*Libbrecht & Noble, 2015*; *Obermeyer & Emanuel, 2016*). To the best of our knowledge, no investigations have used integrated analysis and machine learning methods to identify key IDD-associated genes.

Accumulating evidence has indicated that noncoding RNAs, including microRNAs (miRNAs), long noncoding RNAs (lncRNAs), and circular RNAs (circRNAs), are important gene expression regulators that influence cellular function and disease states (*Adams et al., 2017*). lncRNAs and circRNAs act as competitive endogenous RNAs (ceRNAs) and competitively bind miRNA response elements (MREs) to construct a regulatory network involved in IDD progression. *Zhao et al. (2016)* used RNA sequencing to identify 1,854 lncRNAs and 2,804 protein-coding genes that were differentially expressed in the IDD group. *Tan et al. (2018)* found that LncSNHG1 promoted NP cell proliferation by suppressing miR-326 expression and upregulating *CCND1* expression. Recent studies have focused on the functional roles of circRNAs during IDD development and found that

**Table 1 Basic information of expression profiles included in the study.**

| Type | Series | Platform | Source name | Number of samples (control/degenerative) | Publication year |
|---|---|---|---|---|---|
| mRNA | GSE34095 | GPL96 | Disc tissue | 6(3/3) | 2012 |
| mRNA | GSE15227 | GPL1352 | Disc tissue | 15 (12/3) | 2009 |
| miRNA | GSE63492 | GPL19449 | Nucleus pulposus | 10 (5/5) | 2016 |
| miRNA | GSE116726 | GPL20712 | Nucleus pulposus | 6(3/3) | 2018 |
| mRNA-lncRNA | GSE56081 | GPL15314 | Nucleus pulposus | 10(5/5) | 2014 |
| circRNA | GSE67566 | GPL19978 | Nucleus pulposus | 10(5/5) | 2016 |

circ-4099, circ-GRB10, circVMA21, and circ_001653 play pivotal roles during NP cell proliferation, apoptosis, and extracellular matrix synthesis/degradation (*Cheng et al., 2018*; *Cui & Zhang, 2020*; *Guo et al., 2018*; *Wang et al., 2018a*). The diverse biological functions of ceRNAs deserve further exploration.

In this study, we aimed to identify the key genes and underlying mechanisms of IDD development by constructing an lncRNA/circRNA-miRNA-mRNA network using multiple microarray datasets and machine learning methods. Figure S1 shows the flow chart for this study. Our results present novel biomarkers and therapeutic targets that can be used for IDD diagnosis and treatment.

## MATERIALS & METHODS

### Microarray datasets

We retrieved a total of six expression profiles from the GEO database (http://www.ncbi.nlm.nih.gov/geo): two mRNA expression profiles (GSE34095 (*Tsai et al., 2013*) and GSE15227 (*Gruber et al., 2009*)), two miRNA expression profiles (GSE63492 (*Lan et al., 2016*)) and GSE116726 (*Ji et al., 2018*), one mRNA-lncRNA expression profile (GSE56081 (*Lan et al., 2016*; *Wan et al., 2014*)), and one circRNA expression profile (GSE67566 (*Lan et al., 2016*; *Liu et al., 2015*)). Table 1 contains the basic information for these expression profiles.

### Differential expression analysis

The raw data were annotated, normalized, log 2 transformed, and screened for differentially expressed genes (DEGs), differentially expressed miRNAs (DEMs), differentially expressed lncRNAs (DELs), and differentially expressed circRNAs (DECs) in IDD and normal disc tissue using the "Limma" R package (*Ritchie et al., 2015*). Since each dataset came from a different experiment and microarray platform, the way to get the data may be different. When filtering using consistent thresholds, some datasets did not yield valid differential genes. In order to obtain effective differentially expressed genes for subsequent analysis, we used different thresholds according to each dataset's conditions. The specific DEG screening thresholds were as follows: $p$-value $<0.05$ for the GSE34095 dataset, $p$-value $<0.01$ and $|\log2FC| \geq 2$ for the GSE15227 dataset, and $p$-value $<0.01$ and $|\log2 FC| > 2$ for the GSE116726 dataset. We obtained the GSE67566, GSE63492, and GSE56081 datasets from the same tissue samples, and provided the differential expression analysis results and thresholds as Supplemental Materials (Tables S1 and S2, Documents S1 and S2). We

visualized the differential expression analysis results using a volcano plot and heatmap with hierarchical clustering (Fig. S2).

## Feature gene extraction

Using VENNY 2.1 software (http://bioinfogp.cnb.csic.es/tools/venny/index.html), we extracted the differential gene intersections of the GSE34095 and GSE15227 datasets to use as IDD-related DEGs (Li et al., 2019; Zhang et al., 2018). We used the GSE15227 dataset as the training set to screen for important feature genes. Using the R caret package, random forest, and neural network methods, we constructed the model and obtained the important DEG features (Kuhn, 2015). We used recursive feature elimination (RFE), a machine learning method, to extract the optimum feature genes to identify the functional biomarkers involved in IDD progression (Guyon et al., 2002).

## SVM model verification

The support vector machine (SVM) is particularly effective for binary classification in a supervised learning manner, and is better than other machine learning methods at identifying subtle patterns in complex datasets (Aruna & Dr, 2011). Radial basis function kernel (RBF kernel) is commonly used in nonlinear support vector machine classification since it can enable data to operate in a high-dimensional and implicit feature space (Jiao et al., 2017). In this study, we used the R e1071 package (https://CRAN.R-project.org/package=e1071) to build an RBF kernel SVM model to identify the optimal feature genes, and we selected the GSE15227 dataset to train the machine. We then evaluated its performance in the GSE15227 training set using the R pROC package (http://www.biomedcentral.com/1471-2105/12/77/) (Robin et al., 2011). The optimal feature genes were taken from the training set, and over-fitting would occur whenever ROC verification was performed. Therefore, we used two validation sets (GSE34095 and GSE56081) to further verify the model's performance.

## Identifying target miRNAs of the optimal feature genes

We used miRWalk 2.0 with 12 prediction programs (MIMATid, Microt4, miRanda, mirbridge, miRDB, miRMap, miRNAMap, Pictar2, PITA, RNA22, RNAhybrid, and Targetscan) to predict the optimal feature genes' target miRNAs (Dweep & Gretz, 2015). miRNAs present in more than five of the 12 prediction programs were considered as target miRNAs. We selected overlapping DEMs in the GSE116726 and GSE63492 datasets as candidate DEMs. Intersecting target miRNAs and candidate DEMs were selected as optimal feature gene interaction pairs.

## ceRNA network construction

We obtained the miRNA sequences interacting with optimal feature genes from the miRbase (Kozomara & Griffiths-Jones, 2011) and extracted mature sequences using the Perl program. The DEL sequences were downloaded from NCBI. For DELs with several transcripts, we selected the longest transcripts for subsequent analysis. We used the BEDTOOLS command (Quinlan & Hall, 2010) and genomic coordinates to obtain the DEC sequences, and converted the gene name to its circRNA symbol using the Perl program.

We then used the miRanda tool to analyze the combinations of miRNAs, DELs, and DECs. We set the analysis parameters for the miRNA-DELs and miRNA-DECs as sc:120, en:-20 and sc:150, en:-7, respectively, and processed the results using python script (Documents S3 and S4). The DELs and DECs with ≥5 miRNA binding sites were identified as reliable miRNA-lncRNA and miRNA-circRNA interaction pairs. The ceRNA (DELs/DECs-miRNA-optimal feature gene) regulatory network was constructed using a combination of miRNA-DEL pairs, miRNA-DEC pairs, and miRNA-optimal feature gene pairs. We visualized the network using Cytoscape 3.6.0 (http://www.cytoscape.org/).

## Statistical analysis

All data were presented as the mean ± SEM. Differential expression levels were compared using the Student's *t*-test in GraphPad Prism 7.0 (GraphPad Software Inc., La Jolla, CA, USA). *P* values < 0.05 were considered statistically significant.

# RESULTS

## Differentially expressed genes in IDD

We obtained a total of 334 DEGs (199 up-regulated and 135 down-regulated) in the GSE34095 dataset (Fig. 1A) and 188 DEGs (141 up-regulated and 47 down-regulated) in the GSE15227 dataset (Fig. 1B). Hierarchical cluster heatmaps showed that these DEGs could distinguish between the degenerative disc samples and the control disc samples (Figs. 1C and 1D). We obtained a total of 13 overlapping DEGs (*COL3A1*, *SCRG1*, *HTRA1*, *BCAS4*, *C11orf80*, *CRNKL1*, *GREM1*, *FGFR3*, *BDKRB1*, *WDR46*, *FN1*, *LMF2*, and *GDI2*) via the intersection between the two datasets, and considered these as IDD-related DEGs (Fig. 1E).

## Optimal feature gene selection

We used random forest and neural network models to evaluate the importance of the 13 overlapping DEGs (Documents S5 and S6). The results showed that most of the DEGs from the two models had minimal differences in terms of gene importance rankings (Figs. 2A and 2B). Using the RFE method, we identified four important genes (*WDR46*, *BCAS4*, *CRNKL1*, and *SCRG1*) as optimal feature genes associated with IDD (Fig. 2C).

A classification model was constructed using the RBF kernel SVM, the four genes as features, and the GSE15227 dataset to train the machine. The parameters were set as: SVM-Kernel: radial, cost:1, gamma:0.25, and epsilon:0.1. Additionally, their performance was assessed using the R pROC package. In the GSE15227 training set, the area under the ROC curve (AUC) of the SVM model was 100 percent, suggesting that the model could accurately distinguish between IDD and normal samples (Fig. 2D). We used the GSE34095 and GSE56081 datasets as validation sets to further evaluate the model's performance and to help avoid over-fitting in the training set. The SVM model in the GSE34095 validation set had an AUC of 55.6 percent, which may be correlated with the small sample size of the microarray dataset (Fig. 2E). However, the AUC for the GSE56081 validation set was 100 percent (Fig. 2F). These results indicated that the optimal feature genes (*WDR46*, *BCAS4*, *CRNKL1*, and *SCRG1*) could be used as effective and accurate IDD diagnostic biomarkers.

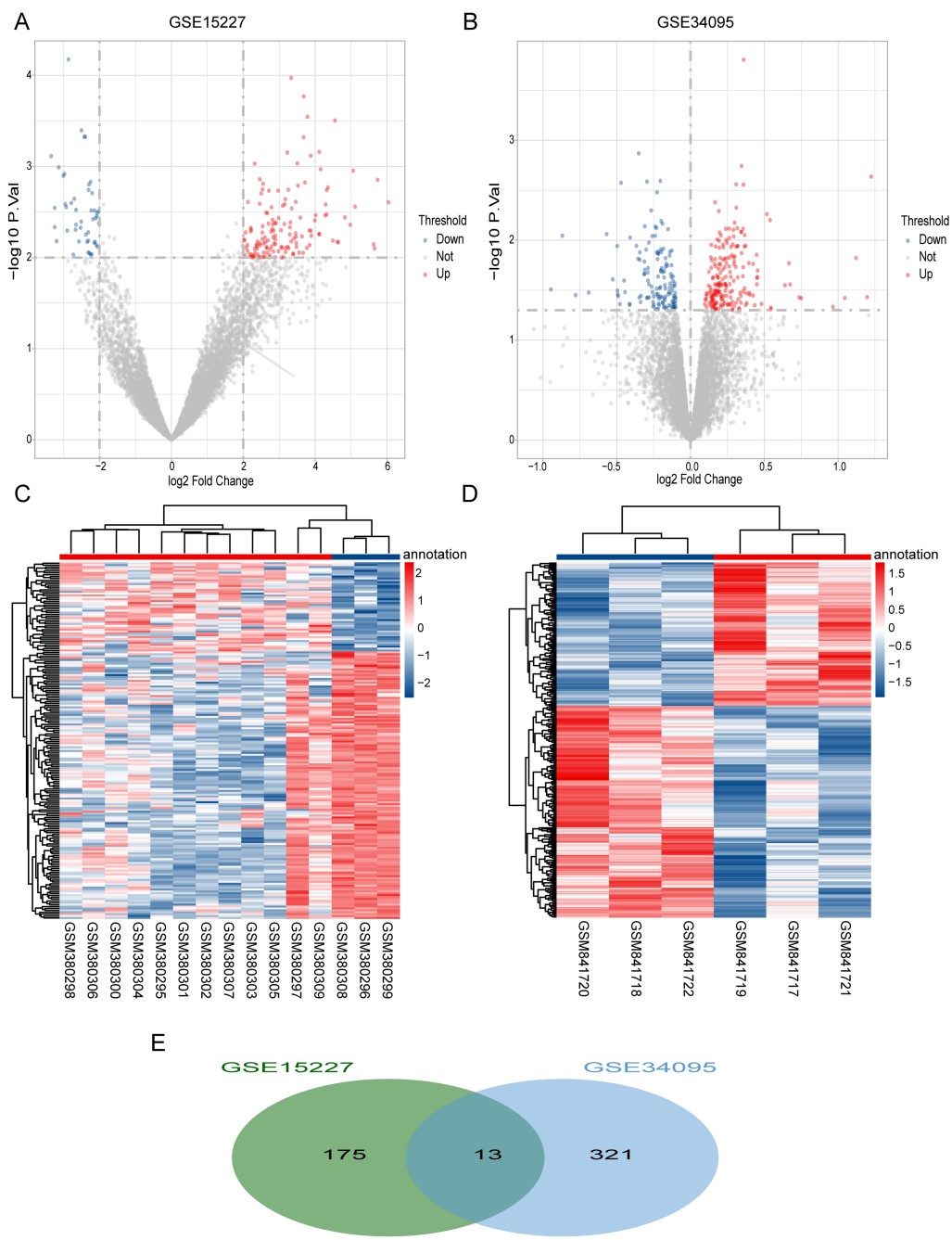

**Figure 1** **Differentially expressed IDD genes.** (A and B) Volcano plots represent the DEGs in the degenerative disc samples and control disc samples in the GSE15227 and GSE34095 datasets, respectively. (C and D) Hierarchical cluster heatmaps of the GSE15227 and GSE34095 datasets displaying the DEGs in the degenerative disc samples and the control disc samples. Blue represents the downregulated and red represents the upregulated. (E) Venn diagram of DEGs in the GSE15227 and GSE34095 datasets. The common area represents the overlapping genes. DEG, differentially expressed genes.

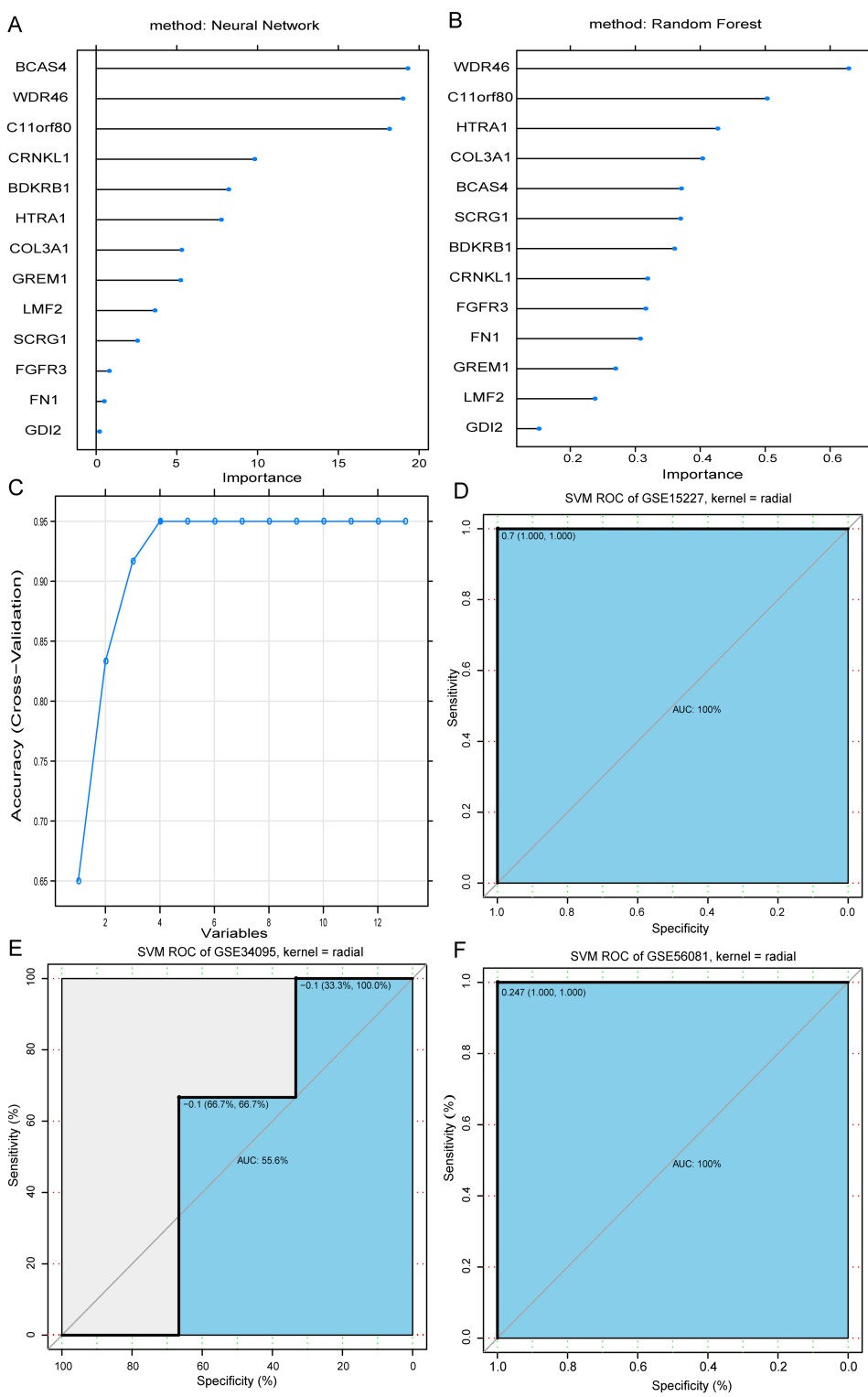

**Figure 2  Selection of optimal feature genes.** Ranking of the top 13 IDD-related genes using neural networks (A) and random forest (B). Extraction of the optimum feature genes from the 13 IDD-related genes was carried out using recursive feature elimination (C). Classification efficiency of the optimum feature genes in the model as evaluated using the ROC curve in the GSE15227 (D), GSE34095 (E), and GSE56081 (F) datasets, respectively.

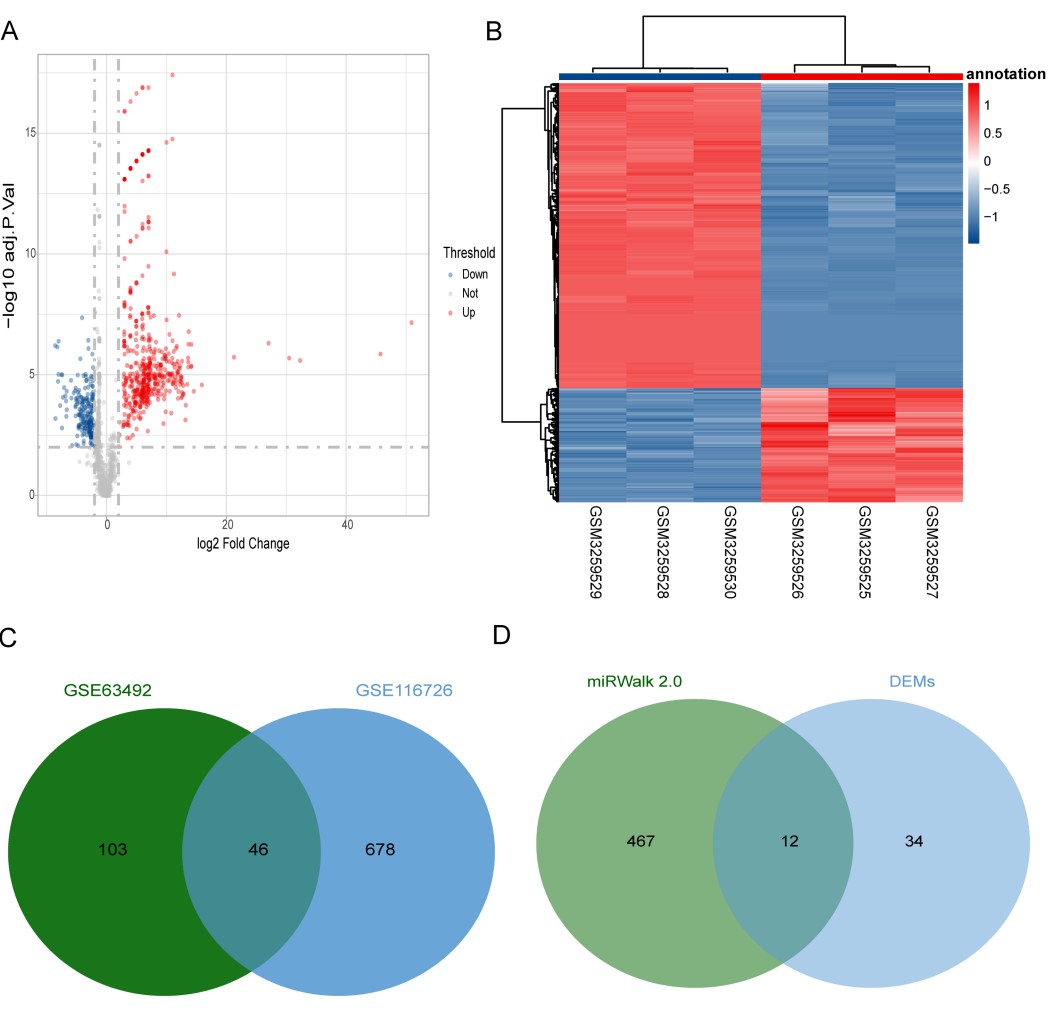

**Figure 3** **Identification of target miRNAs of optimal feature genes.** (A) Volcano plots represent the DEMs of the degenerative disc samples and control disc samples in the GSE116726 dataset. (B) Hierarchical cluster heatmaps of the GSE116726 dataset display the DEMs to compare degenerative disc samples and control disc samples. Blue represents the degenerative samples and red represents the control samples. (C) Venn diagram of DEMs in the GSE116726 and GSE63492 datasets. The common area represents the overlapping DEMs. (D) Venn diagram of miRNAs in overlapping DEMs and target miRNAs of optimum feature genes. The common area represents the overlapping miRNAs. DEM, differentially expressed miRNA.

## Identifying target miRNAs of the optimal feature genes

We predicted a total of 467 target miRNA optimal feature genes using miRWalk 2.0. Compared to the control disc samples, we identified 724 DEMs (527 up-regulated and 197 down-regulated) from the GSE116726 dataset (Figs. 3A and 3B) and 149 DEMs from the GSE63492 dataset. Figure 3C shows a Venn diagram of the 46 common DEMs that were found. Based on the intersections between the target miRNAs and common DEMs, we further analyzed 12 overlapping miRNAs which we considered to be the optimal feature genes' interaction miRNAs (Fig. 3D).

## ceRNA network construction

The differentially expressed lncRNAs and circRNAs between the degenerated disc samples and the control disc samples were downloaded from the GEO database. We analyzed the miRNA-DEL interactions and the miRNA-DEC interactions using the miRanda tool with ≥5 miRNA binding sites. The miRNA-DEL pairs, miRNA-DEC pairs, and miRNA-optimal feature gene pairs were combined to build a DELs/DECs-DEMs-optimal feature gene regulatory network, which included four mRNA nodes, 12 miRNA nodes, 10 lncRNA nodes, and 75 circRNA nodes (Fig. 4, Table S3). Out of these, three miRNAs (hsa-miR-4728-5p, hsa-miR-5196-5p, and hsa-miR-185-5p) and three circRNAs (hsa_circRNA_100723, hsa_circRNA_104471, and hsa_circRNA_100750) were key regulators, based on the optimal feature genes and connective degrees of the whole network (Table 2). However, the LncRNA connective degree was very low.

## Validating optimal feature genes

The expression of the four optimal feature genes across the three datasets was visualized using box plots. In the GSE15227 and GSE56081 datasets, *BCAS4* expression levels were significantly down-regulated (Figs. 5A and 5C), and *SCRG1* was significantly up-regulated (Figs. 5G and 5I). *CRNKL1* was only up-regulated in the GSE34095 dataset (Fig. 5E). Unfortunately, the expression levels of *WDR46* showed no significant difference across the three datasets (Figs. 5J, 5K and 5L). These results indicated that *BCAS4* and *SCRG1* are key genes involved in IDD development.

## DISCUSSION

IDD is linked to lower back pain and spine-related diseases, and although IDD's underlying mechanisms have been studied for many years, they still remain unclear. Insufficient early IDD diagnosis and treatment methods affect the quality of life for patients and impose a heavy economic burden on society (*Vergroesen et al., 2015*). NP cells play an important role in maintaining intervertebral disc homeostasis by synthesizing ECM, which includes aggrecan and type II collagen (*Zhang et al., 2016*). Recent studies show that targeting gene therapy can inhibit NP cell senescence and apoptosis, and can ultimately ameliorate IDD (*Chen et al., 2018*). Therefore, reliable and specific gene targets are essential for IDD diagnosis and treatment.

Recent developments in bioinformatics and computational biology have led to the identification of several key genetic targets related to IDD and the prediction of potential molecular mechanisms (*Petryszak et al., 2014*). In this study, we downloaded multiple microarray datasets associated with IDD from the GEO database, including two mRNA expression profiles, two miRNA expression profiles, one mRNA-lncRNA expression profile, and one circRNA expression profile. A total of four optimal feature genes (*WDR46*, *BCAS4*, *CRNKL1*, and *SCRG1*) were identified using machine-learning methods. The construction of a DELs/DECs-miRNA-optimal feature genes network revealed that three miRNAs (hsa-miR-4728-5p, hsa-miR-5196-5p, and hsa-miR-185-5p) and three circRNAs (hsa_circRNA_100723, hsa_circRNA_104471, and hsa_circRNA_100750) may be important mediators for optimal feature genes. To further explore the different optimal
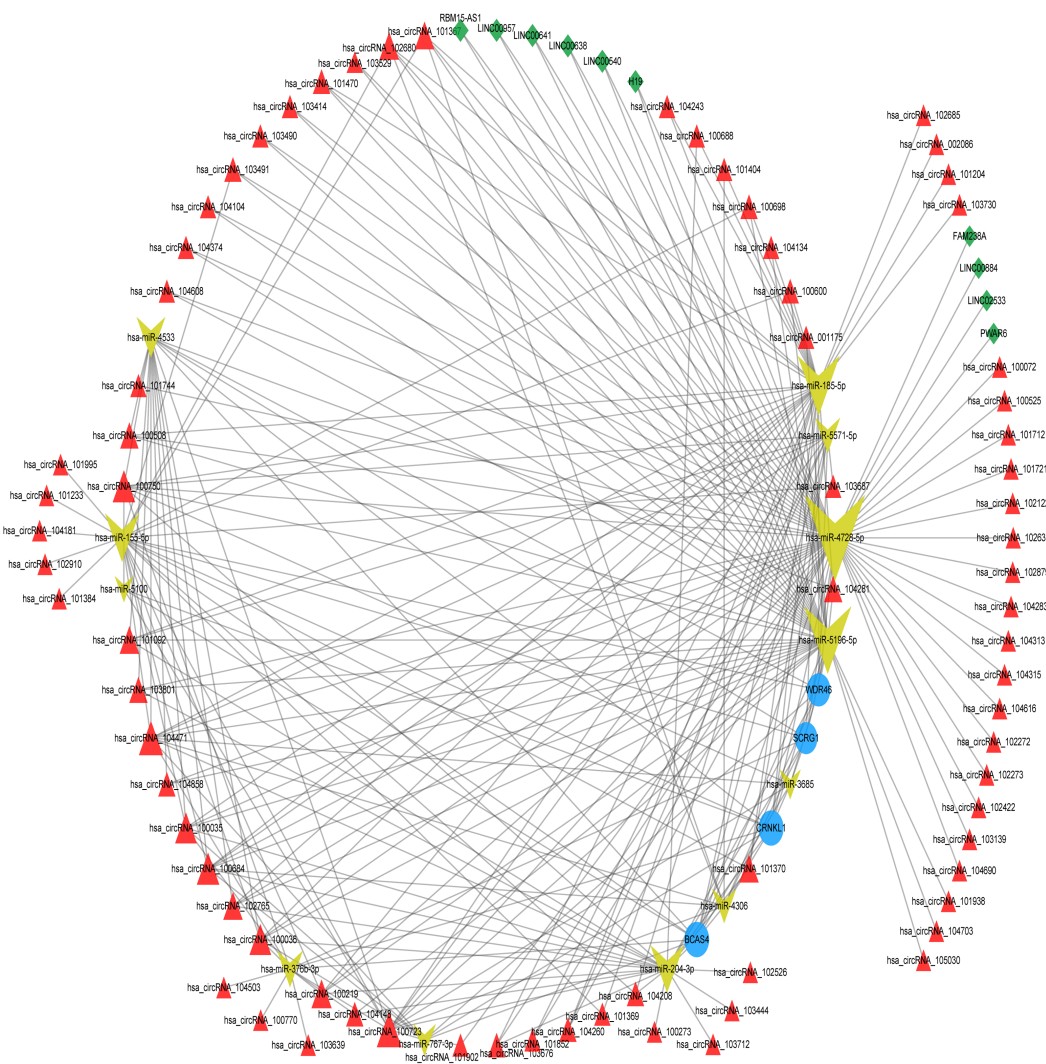

**Figure 4   ceRNA network construction.** ceRNA network of optimum IDD feature genes with DEMs, DELs, and DECs. The blue circles represent optimum IDD feature genes, the red triangles represent DECs, the green diamonds represent DELs, and the yellow arrows represent DEMs. ceRNA, competing endogenous RNA; DEC, differentially expressed circRNA; DEL, differentially expressed lncRNA; and DEM, differentially expressed miRNA.

feature genes in normal and degenerative NP tissues, we also investigated their expression levels across three datasets. The results indicated that *BCAS4* and *SCRG1* were key genes related to IDD.

Breast carcinoma amplified sequence 4 (*BCAS4*), a novel gene cloned from breast cancer cells, encodes a 211-amino acid cytoplasmic protein with no significant homologies to any known protein (*Barlund et al., 2002*). Previous studies have demonstrated that the specific DNA methylation of *BCAS4* acts as an epigenetic marker and can be used to distinguish saliva from other body fluids. It is also widely used in forensic investigations (*Silva et al., 2016*; *Taki & Kibayashi, 2015*). However, *BCAS4*'s biological role in disease requires

**Table 2   The top 3 miRNAs and circRNAs related to optimal feature genes in the network.**

| Type | Name | Number of directed edges |
| --- | --- | --- |
| miRNA | hsa-miR-4728-5p | 58 |
| miRNA | hsa-miR-5196-5p | 41 |
| miRNA | hsa-miR-185-5p | 34 |
| circRNA | hsa_circRNA_100723 | 12 |
| circRNA | hsa_circRNA_104471 | 11 |
| circRNA | hsa_circRNA_100750 | 9 |

further investigation. Stimulator of chondrogenesis 1 (*SCRG1*) was first found in the genes associated with, or responsible for, the neurodegenerative changes observed in transmissible spongiform encephalopathies (*Dandoy-Dron et al., 1998*). *SCRG1* transcript is found in the brain, heart, and spinal cord, and its sequence is highly conserved in humans, mice, and rats. *SCRG1* has also been observed to be specifically expressed in human articular cartilage, and is involved in human mesenchymal stem cell (hMSC) growth suppression and differentiation during dexamethasone-dependent chondrogenesis (*Ochi, Derfoul & Tuan, 2006*). Recent studies have shown that *SCRG1* is an important regulator during hMSC self-renewal, migration, and osteogenic differentiation along with its receptor *BST1* (*Aomatsu et al., 2014*). However, the function of *SCRG1* in IDD development has not yet been explored.

circRNAs and lncRNAs may act as ceRNAs by competitively binding to miRNA and suppressing mRNA expression (*Adams et al., 2017*; *Tay, Rinn & Pandolfi, 2014*). This ceRNA hypothesis suggests that there is a novel mechanism for RNA interactions. In this study's ceRNA analysis, hsa-miR-4728-5p, hsa-miR-5196-5p, hsa-miR-185-5p, hsa_circRNA_100723, hsa_circRNA_104471, and hsa_circRNA_100750 showed more interactions compared to the other RNAs in the whole network. Additionally, miR-155, miR-21, and miR-133a were shown to be differentially expressed in degenerative NP cells, indicating that they may be potential biomarkers for early IDD diagnosis (*Liu et al., 2014*; *Wang et al., 2011*; *Xu et al., 2016*). The dysregulation of these miRNAs is closely associated with NP cell apoptosis which affects IDD progression. miR-185-5p has been reported as a critical regulator, but miR-4728-5p and miR-5196-5p have rarely been reported. Chang et al. (2017) reported that miR-185-5p induced by *Runx2* could directly target *Dlx2* to inhibit amylogenesis and osteogenesis, providing a new treatment option for cleidocranial dysplasia. Multiple lncRNAs bind to miR-185-5p in order to modulate the progression of different types of human cancer, including prostate cancer (*Tian et al., 2018*), colorectal cancer (*Zhu et al., 2018*), and glioblastoma (*Wang et al., 2018b*). However, the regulatory effects of miR-185-5p on IDD require further investigation.

CircRNAs, a new type of ncRNAs formed by special loop splicing, are thought to be potential diagnostic biomarkers and therapeutic targets because they are more stable and conserved than other RNAs (*Lee et al., 2019*). However, very few studies have focused on the role of circRNAs in IDD development. *Guo et al. (2018)* found that circ-GRB10 was downregulated in IDD cells, and circ-GRB10 overexpression inhibited NP cell apoptosis by

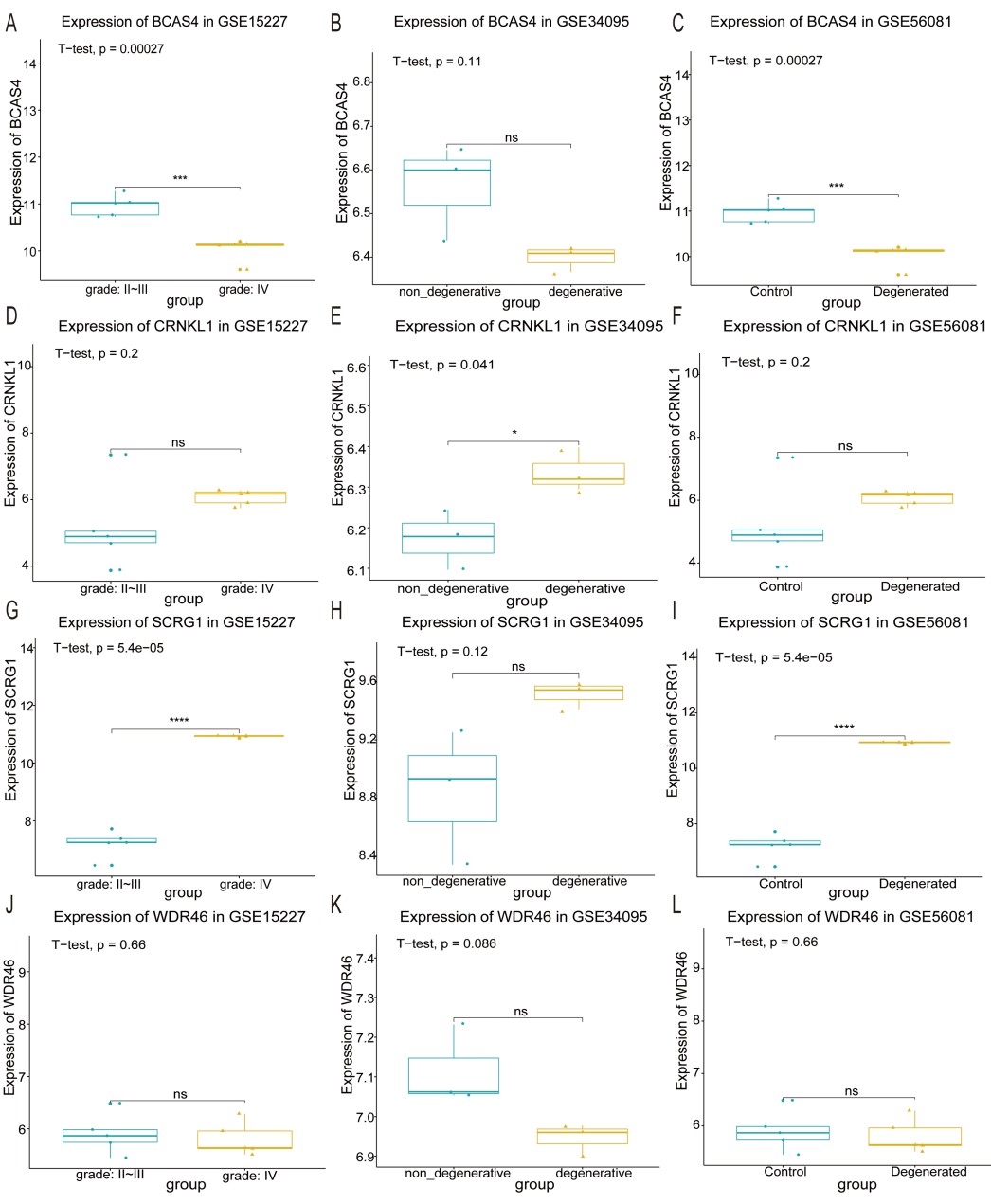

**Figure 5   Validation of optimal feature genes.** The expression levels of the four optimum feature genes in the GSE15227, GSE34095, and GSE56081 datasets, respectively. (A, B and C) The expression level of BCSA4. (D, E and F) The expression level of CRNKL1. (G, H and I) The expression level of SCRG1. (J, K and L) The expression level of WDR46. * represents $P$ value < 0.05, *** represents $P$ value < 0.001, **** represents $P$ value < 0.0001, and NS represents not significant.

sequestering miR-328-5p and upregulating target genes involved in cell proliferation through the ErbB pathway. *Cheng et al. (2018)* reported that circVMA21 acted as a sponge for miR-200c and regulated the activity and function of NP cells by targeting miRNA-200c and *XIAP*, providing a new IDD intervention and treatment strategy.

*Wang et al. (2018a)* found that circRNA_4099 could act as a sponge for miR-616-5p and eliminate *Sox9* inhibition, increasing ECM secretion. Similarly, *Cui & Zhang (2020)* reported that circ_001653 silencing may bind to miR-486-3p in order to inhibit *CEMIP* expression, thus attenuating NP cell apoptosis and ECM degradation. In this study, we used integrated analysis to identify that hsa_circRNA_100723, hsa_circRNA_104471, and hsa_circRNA_100750, which have not been previously reported, are more likely to be important molecules involved in IDD regulation, and require further investigation.

## CONCLUSION

In this study, we used integrated bioinformatics analysis and machine learning methods to identify *BCAS4* and *SCRG1* as key genes associated with IDD development. Additionally, after constructing the ceRNA network, we found three miRNAs and three circRNAs that may act as important regulators during IDD development by targeting key genes. This novel study may provide new insights into IDD pathogenesis and therapy. Further experiments should be conducted to verify this study's results.

### Funding
This work was supported by the Shanghai Natural Science Foundation (NO.20ZR1452400). The funders had no role in study design, data collection and analysis, decision to publish, or preparation of the manuscript.

### Grant Disclosures
The following grant information was disclosed by the authors:
Shanghai Natural Science Foundation: 20ZR1452400.

### Competing Interests
The authors declare there are no competing interests.

### Author Contributions
- Hongze Chang performed the experiments, analyzed the data, authored or reviewed drafts of the paper, and approved the final draft.
- Xiaolong Yang, Kemin You and Mingwei Jiang performed the experiments, prepared figures and/or tables, and approved the final draft.
- Feng Cai, Yan Zhang, Liang Liu and Hui Liu analyzed the data, prepared figures and/or tables, and approved the final draft.
- Xiaodong Liu conceived and designed the experiments, authored or reviewed drafts of the paper, and approved the final draft.

### Data Availability
Raw data is available in the Supplemental Files. The datasets are available from Gene Expression Omnibus (GEO): GSE34095, GSE15227, GSE63492, GSE116726, GSE56081 and GSE67566.

## Supplemental Information

Supplemental information for this article can be found online at http://dx.doi.org/10.7717/peerj.10120#supplemental-information.

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
