# Peer review of "Integrating multiple microarray dataset analysis and machine learning methods to reveal the key genes and regulatory mechanisms underlying human intervertebral disc degeneration"

_PeerJ, doi:10.7717/peerj.10120_

## Round 0.1 · original submission · Major Revisions

Hi Dr. Liu,
Thank you for your submission to PeerJ. Attached are the comments from the reviewers. The reviewers believe your research is valuable. I encourage you to revise the manuscript accordingly and submit it again.

Thanks!
Jianye

·

Basic reporting

1. Review by a primary English writer for grammar, English composition. Several mistakes should be corrected in the manuscript. Such as:
Page 6: 69-72 “Diagnosis of degenerative disc disease is difficult, and currently available treatment and management strategies for IDD primarily involve conservative or surgical treatment to relieve pain, without resolving underlying tissue pathology(An et al. 2003; Zaina et al. 2016).” “involve” maybe not suit in the present context, “consist of” may be more appropriate.
Page 6: 72-73 “Therefore, a detailed understanding of the molecular and cellular events that underlie the formation of IDD is needed to identify diagnostic markers and design new therapeutic targets.” This sentence needs to be rewritten. “to” is not suited in the sentence.
Page 10: 239-243 “Recent developments in bioinformatics and computational Biology has led to the identification of several key genetic targets related to IDD and potential molecular mechanisms have also been predicted (Petryszak et al.2014). In this study, integrated analysis of multi-microarray datasets including two mRNA expression profiles, two miRNA expression profiles, one mRNA-lncRNA expression profile, and a circRNA expression profile, associated with IDD was downloaded from the GEO database.” “Biology” should be corrected as biology. “multi-microarray datasets” is incorrect in English. The sentence, “In this study, integrated analysis of multi-microarray datasets including two mRNA expression profiles, two miRNA expression profiles, one mRNA-lncRNA expression profile, and a circRNA expression profile, associated with IDD was downloaded from the GEO database.” should be rewritten.
Page 11: 266-267 The sentence, “The ceRNA hypothesis is a new gene regulatory model that plays an important role in various diseases.”, should be rewritten.
Page 12: 294-295 “In conclusion, in this study, BCAS4 and SCRG1as were identified as key genes associated with the progression of IDD using bioinformatics analysis and machine-learning methods.” “as” should be deleted.
Page 12: 295-296 The sentence, “Besides, 3 miRNAs and 3 circRNAs were shown to play a vital role in IDD through interaction with key genes after constructing a ceRNA network.”, should be rewritten.
2. Page 6: 67-69 “Previous studies have indicated that nucleus pulposus (NP) cell apoptosis (Jiang et al. 2013), and several other factors including genetic and environmental factors are the main causes of IDD (Battie et al. 2008)”.
Genetic and environmental factors are the primary influencing factors of IDD. Nucleus pulposus cell apoptosis is a phenomenon and process observed in the degeneration, not a cause of IDD. This statement is not accurate.
3. Page 7: 109-110 “DEGs, miRNA(DEMs), lncRNAs (DELs), and circRNAs (DECs) in IDD compared to normal discs tissues were identified using the “Limma” R package (Ritchie et al. 2015).”
The abbreviations in the sentence should be normalized.
4. Page 10: 234-235 “NP plays an important role in maintaining the intervertebral disc homeostasis by synthesizing aggrecan, type II collagen, and extracellular matrix (ECM).”
Aggrecan and type II collagen are the components of ECM. This statement is not accurate.
5. Page 6: 75-97 “Genetic factors play a significant role in the development of IDD, and variants in several genes have been identified (Battie et al. 2008). Studies have reported that vitamin D receptor (VDR), COL1A1, and COL9a3 genes are associated with the degeneration of the lumbar disc, and this is significantly more pronounced in individuals with multiple mutations (Toktas et al. 2015) . In another study, bioinformatics analyses revealed that five genes (FYN, PRKCD, YWHAB, YWHAZ, and AR) were associated with IDD (Ji et al. 2015). However, the relationship between genes and IDD remains controversial and there is a need for further studies.”
Page 6: 88-92 “Tan et al. suggested that LncSNHG1 promotes NP cell proliferation by suppressing miR-326 expression and upregulating CCND1 expression (Tan et al. 2018). Wang et al. demonstrated that circRNA_4099 can sponge to miR-616-5p and eliminate the inhibition of Sox9 by miR-616-5p hence increasing the secretion of the extracellular matrix (Wang et al. 2018). However, the regulatory mechanism of the lncRNA/circRNA-miRNA-mRNA network in the IDD has not been explored.”
In the introduction segment, the authors should provide a greater perspective on the number of studies that have addressed this same research question, analyze the unsolved question exist in this aspect, and expound why this research question is urgent and meaningful.

Experimental design

6. Page 7: 110-115 “Different thresholds were used according to the conditions of each dataset. The specific screening thresholds for DEGs were as follows: GSE34095 dataset was p-value< 0.05; GSE15227 dataset was p-value< 0.01 & |log2FC|≥2; GSE116726 dataset was p-value< 0.01& |log2 FC|>2. The GSE67566, GSE63492, and GSE56081 datasets were obtained from the same tissue samples, and the differential expression analysis results and thresholds were uploaded in the raw data (Table S1,S2;Document S1,S2).”
P-value< 0.05 and |log2FC|≥2 is the generally accepted screening thresholds for the differential expressed genes. In the authors’ study, different thresholds were set for several data sets. Context should be added to further explain the rationale for the setting of different thresholds.

Validity of the findings

7. Page 11: 257-262 “SCRG1 (scrapie responsive gene 1) was first discovered in the brain of scrapie-infected mice (Dandoy-Dron et al. 1998) . Recent studies have shown that SCRG1 is closely related to neurodegeneration, and is an important regulator of human mesenchymal stem cells (hMSCs) self-renewal, migration and osteogenic differentiation with its receptor BST1 (Aomatsu et al. 2014; Dron et al. 2006) . SCRG1 gene encodes a 98-amino acid, a cytokine-like protein that is highly conserved in mammals and has no significant homology to any other known protein(Dandoy-Dron et al. 2003; Dron et al. 2000) .”
SCRG1 is a member of human genes, the Ensemble Gene ID of which is ENSG00000164106. The description of SCRG1 in Ensemble is “stimulator of chondrogenesis 1”. Scrg1 is a member of the mouse genetic base, the description of which in Ensemble is “scrapie responsive gene 1”, with the gene ID of ENSMUSG00000031610. The discussion should be expanded about the SCRG1, not Scrg1.
8. There are several statements and recommendations in the discussion section that are not supported by this study. The authors are encouraged to remove biased language and hyperbole throughout the manuscript. such as :
page 11: 262-263 “The high conservation suggests that BCAS4 and SCRG1 can serve as good diagnostic markers of IDD.”
page 11: 278-279 “Therefore, based on these previous findings, this study presents new insights into the molecular mechanism of IDD development.”
page 12: 296-297 “These are reliable and effective molecular targets for the diagnosis and treatment of IDD.”
9. The tissue samples used in the six data sets were different. While GSE34095 and GSE15227 are the disc tissue, the other four are the NP tissue. The analysis across the six data sets may led to a violated validation of results.
The authors are encouraged to consider discussing the limitations of the study, including but not limited to the item mentioned above.

Additional comments

The authors presented an integrated analysis of several microarray datasets. Bioinformatics and machine learning methods were employed in the analysis. Several genes, miRNAs, and circRNAs were revealed as the key regulators involved in the degenerated progress. Crosstalk among mRNAs, miRNAs, lncRNAs, and circRNAs in IDD was displayed in the paper. The research perspective is very novel. I appreciate their eagerness to provide insights into this area.

·

Basic reporting

The paper is devoted to a meta-analysis of publicly available expression datasets for intervertebral disc degeneration by machine learning techniques. The paper is mostly well written, well referenced and properly organised. Figures and tables are of good quality. All the results are clearly presented and discussed. There are some minor inaccuracies in the typing, namely spaces between end of sentences and full stops, e.g. "(Dandoy-Dron et al. 1998) ."

Experimental design

The design of the study is sound, the study pipeline is clear, the methods are adequate. The study hypothesis is well defined. Methods are fully described and the study seems reproducible. A set of various ML methods was applied with cross-validation between them, thus increasing validity of the findings. This type of studies in rare in the field, therefore, must be encouraged.

Validity of the findings

Findings seem to be meaningful, some of the genes identified are known to be involved in IDD from other studies, thus proving the validity of the study. Any possible bias may be due to the use of publicly available datasets that can be biased one way or another. Discussion is relevant. All the raw and summary data is provided.

Additional comments

I have no specific issues with this study, it seems to be well thought and implemented. There are some minor issues to be addressed:
1. Make sure all gene names are in italic.
2. In introduction, you may want to mention the results of GWASs on IDD.
3. line 123: of the DEGs (Kuhn 2015). Recursive feature elimination (RFE), which is a machine-learning method, was used - which is a machine-learning method phrase is obsolete here
4. line 187: The parameters of the SVM model are shown in Fig.S3. In - just list the parameters in the text, no need for this figure
5. lines 201-202: including physical interactions, co-expression, predicted, co-localization, pathway, genetic interactions and shared protein domains, accounted for 67.64%, 13.50%, 6.35%, 6.17%, 4.35%, 1.40% and 0.59%, respectively. - it is unclear what this percentages represent, account for what? This needs to be rephrased.
6. Line 239: Biology -> biology
7. line 294: "In conclusion, in this study, BCAS4 and SCRG1as were identified as key genes associated with the progression" - you cannot claim these genes are associated with the progression of the IDD as you analysed cross-sectional data; to talk about progression you need to analysed longitudinal data. I would rephrased "key genes associated with the development".

·

Basic reporting

The manuscript presents a valuable analysis of biomarker genes relevant to intervertebral disc degeneration (IDD). The topic itself is of importance since the gene signatures shed light on diagnosis and treatment of IDD. The paper is in general well organized. It is however not easy to follow – this is to some degree expected, given that many datasets and many bioinformatics approaches were introduced. I would suggest add more detailed and concrete explanations, especially detailed description of relationship between non-coding RNA and genes. The authors didn’t cite enough sources to provide background about this study: 1) several studies have been done to identify potential biomarkers for IDD using microarray data (DOI: 10.3892/mmr.2017.7741, https://www.scielo.br/pdf/gmb/v36n3/a21v36n3.pdf); 2) Since RNA-Seq has emerged as an alternative method for gene expression profiling, the results from RNA-Seq can be cited (i.e. https://doi.org/10.1155/2016/3684875). The workflow in Fig. S1 is very helpful for understanding experimental design in this study, but it should be well organized (i.e. all GEO dataset should be placed on the top; and arrow line shouldn’t cross boxes).

Experimental design

The topic itself fall within the scope of PeerJ and will be of interests to the readers of the journal. However, the aim and purpose of the study was not clearly presented: 1) why integrating multiple microarray datasets can improve identification of biomarkers; 2) what roles machine learning methods play in this study?. In Line 112, the different thresholds (p-value < 0.05 & p-value < 0.01) were used and the authors didn’t explain the reason why using different p-values. In “SVM model verification”, the authors should include more details about SVM models, such as kernel (it is very important, so don’t just put R output in Fig S3). I am wondering how the training dataset(s) and validation dataset(s) were decided. Also, there are different machine learning methods (random forest, K-nearest neighbors, decision tree etc.) which can be used for verification. So, the authors can apply these methods and do comparison.

Validity of the findings

Since several studies have been done for identification of biomarkers for IDD, the authors need to claim the novelty of this study. Since only 13 genes were found overlap between DEGs from GSE34095 and GSE15227, they could be overlapped by chance (False positives). So, the authors should implement overlap test (i.e. hypergeometric test). In Line 192, I don’t think the sample size is reason of low AUC for GSE34095. In this study, BCAS4 and SCRG1 were identified as final key genes, however CRNKL1 was used to predict the relevant genes in Line 198 and 199. It doesn’t make sense.

Additional comments

The manuscript presents a valuable analysis of biomarker genes relevant to intervertebral disc degeneration (IDD). The topic itself is of importance since the gene signatures shed light on diagnosis and treatment of IDD. I would suggest add more detailed and concrete explanations.

---

## Round 0.2 · Minor Revisions

Dear Dr. Liu,

One reviewer suggested some minor revisions to your updated manuscript.

Thanks!
Jianye

·

Basic reporting

The authors have addressed all of concerns from my previous review.

Experimental design

The authors have revised the manuscript substantially and addressed most of issues raised. The explanation about using different p-values for different experiment and microarray platform are not clear to me. I am wondering if the similar numbers of DEGs were generated from different experiments when using different p-values? I recommend that the authors add one or two more sentences to explain it.

Validity of the findings

The authors have thoroughly addressed my comments. The authors mentioned " Moreover, many studies have used the method of intersecting multiple datasets to further determine the differential expression genes." I recommend that the authors cite some references to them.

Additional comments

The manuscript is, in my opinion, ready to be published in PeerJ with minor corrections and clarifications.

---

## Round 0.3 · accepted · Accept

Thank you so much for the quick revision. The manuscript now is ready for publication.